# Farmers’ Market Nutrition Program Educational Events Are Broadly Accepted and May Increase Knowledge, Self-Efficacy and Behavioral Intentions

**DOI:** 10.3390/nu14030436

**Published:** 2022-01-19

**Authors:** Karla L. Hanson, Xiangqi Meng, Leah C. Volpe, Stephanie Jilcott Pitts, Yvonne Bravo, Jennifer Tiffany, Rebecca A. Seguin-Fowler

**Affiliations:** 1Department of Public and Ecosystem Health, Cornell University, Ithaca, NY 14853, USA; xm93@cornell.edu (X.M.); lmc267@cornell.edu (L.C.V.); 2Department of Public Health, East Carolina University, Greenville, NC 27858, USA; jilcotts@ecu.edu; 3Cornell University Cooperative Extension—New York City Programs, New York, NY 10022, USA; yb53@cornell.edu (Y.B.); jst5@cornell.edu (J.T.); 4Texas A&M AgriLife Research, College Station, TX 77843, USA; r.seguin-fowler@ag.tamu.edu

**Keywords:** farmers’ markets, nutrition education, food shopping pattern, cooking demonstration, food sampling, fruit and vegetable intake

## Abstract

The Farmers’ Market Nutrition Program (FMNP) in the U.S. provides coupons for the purchase of fruit and vegetables (FV) to pregnant women and children enrolled in the Special Supplemental Nutrition Program for Women, Infants and Children (WIC), and to income-eligible adults 60+ years of age. The New York State FMNP Education Event Guide was developed to support cooperative extension educators in providing information, food tastings, and cooking demonstrations at farmers’ markets (FM) to encourage consumption of FV. This paper describes implementation at seven FM in New York City, and shopping and eating behaviors in a cross-sectional survey of FM shoppers (*n* = 377). Three of nine lessons were implemented more than once, typically with food sampling (78.9%). FM shoppers were primarily women (81.5%), racially diverse (30.5% Black, 23.1% White), frequent shoppers (2.4 times/month), and had high FV consumption (2.24 cups fruit; 2.44 cups vegetables daily). Most FM shoppers participated in the FM education event (84%), and participants and non-participants had equivalent shopping and eating behaviors. More than 70% of FM education participants believed that the event positively impacted their knowledge, self-efficacy, and behavioral intentions. FMNP education events at FM were broadly accepted by FM shoppers of all characteristics, and may improve knowledge, self-efficacy, and behavioral intention.

## 1. Introduction

Fruit and vegetable (FV) consumption can reduce the risk of cardiovascular disease, type-two diabetes, and some cancers [1,2,3,4]. However, only one in ten U.S. adults eats the recommended quantities of FV [5]. Furthermore, FV intake includes little variety: Half of total vegetable consumption is potatoes and tomatoes, and dark green and orange vegetables are seldom consumed [5]. Consumption of FV is even lower among adults in low-income [6] and food insecure households [7]. Adults in low-income households face barriers to FV access including lack of knowledge about the benefits of FV, unhealthy food environments, and a lack of time to cook [8,9,10]. Access to farmers’ markets is one strategy to improve healthy eating behaviors such as FV consumption [11,12]. Prior studies provide consistent evidence that shopping at a FM at least once in the past year was associated with higher reported fruit [13] or FV intake [14] and greater likelihood of consuming five or more FV daily [15]. Furthermore, more frequent FM shopping was associated with more frequent FV consumption [16] as well as higher reported [17,18,19] and objectively measured [19] FV intake.

The Farmers Market Nutrition Program (FMNP) is a U.S. program to support use of FMs by residents with financial barriers to access. FMNP provides coupons for the purchase of FV to pregnant women and parents of children enrolled in the Special Supplemental Nutrition Program for Women, Infants and Children (WIC), and to adults 60+ years of age who meet similar low-income guidelines [20]. In FY 2020, nearly 1.2 million WIC participants [21] and more than 725,000 seniors [22] received FMNP coupons. FMNP benefits range from USD 10 to USD 30 per year for WIC enrollees [21] and USD 20 to USD 50 per year for seniors [22] but can be supplemented by state or local funds in some states [20]. In fiscal year 2020, more than 15,000 farmers, more than 2000 FMs, and more than 2000 roadside stands were authorized to accept FMNP coupons as payment [20].

In most states, FMNP is supported by local cooperative extension educators who provide educational programming to help shoppers learn about healthy eating, try new FV, and develop skills and confidence to select, store, and prepare FV. There is some evidence that nutrition education can change attitudes and behaviors, which may lead to increases in the frequency, types, and quantity of FV consumption [23,24]. For example, in one study, participants who completed two or more FM-based nutrition education classes consumed one-half cup more FV and held more positive attitudes toward trying new FV compared to those who completed only one or zero classes [23]. Cooking demonstrations and tasting activities have also been suggested as ways to improve eating habits and diet quality by reducing food neophobia and fostering positive food experiences [25]. However, results are mixed. One study reported that hands-on participation in food preparation and meal sharing were associated with improved FV consumption, self-efficacy, and nutritional knowledge [26], whereas another study reported that cooking demonstrations and food tastings were not associated with behavior change [27]. Therefore, more research is needed to better understand whether nutrition education and demonstrations at FMs can increase FV consumption and improve dietary quality.

This paper presents an evaluation of an FMNP Education Event Guide developed through a collaboration between Cornell University and Cornell Cooperative Extension to support cooperative extension educators in delivering such events. The Guide consists of nine lessons for delivery at FMs from which extension educators select for delivery at each market event. Each lesson was accompanied by information on food assistance programs that could be used at the FM and included a brochure explaining each program. Extension educators were encouraged to incorporate food sampling and/or cooking demonstrations into their lessons, during which they provided information on how to best select, store, prepare, and cook the featured produce. In the summer of 2019, researchers at Cornell University oversaw implementation of events using The Guide by Cornell Cooperative Extension (CCE), with funding provided by the New York State (NYS) Department of Agriculture and Markets.

This study had three objectives. First, we aimed to better understand which FM shoppers participated in FMNP education events by comparing the socio-demographic characteristics of participants and non-participant shoppers. Although any shopper may participate in FMNP education events, we anticipated that familiarity with the FMNP program and logo may result in shoppers who receive FMNP coupons being more likely to participate in FMNP education activities. Therefore, given the FMNP eligibility criteria, we hypothesized that participants will be more likely to be older, to receive WIC benefits, and to have lower incomes relative to non-participant shoppers. Second, we compared the shopping and eating behaviors of education event participants and non-participant shoppers. Given prior research, we hypothesized that participation in FMNP education activities would be positively associated with shopping and eating behaviors and FV consumption. Third, we described participants’ perceptions of the impact of FMNP education on their knowledge, self-efficacy, and intended behavior. We hypothesized that participants would perceive that the FMNP education event led to increased knowledge about the importance of healthy eating and the preparation and storage of FV; improved self-efficacy to select and purchase FV; and increased acceptance of unfamiliar FV and ability to prepare quick and healthful FV recipes.

## 2. Materials and Methods

This study utilized cross-sectional data collected at FM in New York City. First, researchers observed and recorded information about the FM and the FMNP education event. Second, FM shoppers completed a survey that asked about their shopping and eating behaviors and participation in the FMNP educational event, and participants were asked to rate statements related to the impact of the event on their knowledge, self-efficacy, and behavioral intention. A sample size target of 240 respondents was calculated to yield statistical power to detect moderate differences in vegetable intake between education event participants and non-participant shoppers (α = 0.05, β = 0.80, and effect size = 0.5 SDs), and that target was exceeded.

Seven FM in New York City were visited a total of 26 days in July and August 2019. These FM were chosen in collaboration with staff at GrowNYC because they were located next to a WIC clinic or a senior center (which made it easier to participate in nutrition education and purchase produce using FMNP coupons), in neighborhoods in which housing costs and incomes were lower (to reach shoppers who might use SNAP benefits to purchase FV), no nutrition education was planned by other agencies, and water and storage were available for cooking demonstrations.

### 2.1. Data Collection Approach and Measures

Observational data were collected by researchers through a market audit form adapted from the validated Farmers’ Market Audit Tool [28] implemented in a prior study [29]. Data included the number of FM vendors and number of vendors who sold FV. Researchers also observed the FMNP Educational Event and recorded which of the lessons was being presented, and whether the presentation included a cooking demonstration, food sampling, both activities, or neither. For lessons that included cooking demonstrations or sampling, researchers also recorded the name of the recipe or sample, as well as which FV were ‘featured’. Researchers also recorded whether non-FMNP activities such as nutrition education, cooking demonstrations, samplings, or market tours occurred at the FM on that day.

English-speaking FM shoppers at least 18 years of age were eligible to participate in the survey regardless of whether they participated in FMNP education or not. Public intercept recruitment was used by research staff who explained the purpose of study and obtained verbal consent. Surveys could be completed on an electronic tablet or paper (which were subsequently entered into the electronic survey by research staff). For all constructs of interest, short and focused tools were selected resulting in a survey that lasted only 5–10 min.

The FM shopper survey collected information on socio-demographic characteristics, FM shopping behaviors, FV intake, and perceived impacts of the education event. Socio-demographic characteristics included age (6 categories), gender (male, female, other/prefer not to answer), marital status (6 categories), race (seven options), ethnicity (Hispanic or not), and income level (7 categories). Respondents were also asked if they currently received WIC benefits, and whether they currently received benefits from the Supplemental Nutrition Assistance Program (SNAP, or SNAP/EBT) or food stamps.

Usual and survey day FM shopping behaviors were assessed on eight dimensions. First, respondents were asked as “During the FM season, approximately how often do you purchase fruits or vegetables from the FM?”, which was adapted from a U.S. surveillance survey and implemented in prior research [17]. Five response choices were offered, from “Once a week” to “Never”, which were subsequently transformed into monthly frequencies. Spending was assessed as: “When you go to a FM, how much money (cash and/or benefits) do you usually spend on fresh fruits and vegetables?” with respondents reporting whole dollar amounts [17]. Respondents who indicated that they were currently enrolled in WIC and those who were 60 years or older were asked “Have you ever used Farmers’ Market Nutrition Program (FMNP) coupons at this or another farmers’ market?”. Respondents who indicated that they currently received SNAP benefits were asked “Have you ever used SNAP/EBT or food stamps benefits at this or another farmer’s market?”. Respondents also reported whether they planned to purchase specific FV before they came to the market on the survey day (yes or no) and were asked to list the FV they planned to buy in open text. Respondents were also asked whether they purchased any FV on the survey day (yes or no) and to indicate which FV they purchased on a checklist of 19 fruits and 50 vegetables commonly grown based on a New York State agricultural calendar [30] and a Northeast Regional Food Guide [31]. For respondents that shopped at a FM that ‘featured’ a produce item in a lesson, food sampling, or cooking demonstration, purchased FV were subsequently compared to the featured FV to create an indicator of whether the respondent “purchased the featured FV”. Respondents were also asked “Are you done shopping at the market today?”.

FV intake was assessed by self-report and objective measurement of skin carotenoids, which are a biomarker of FV intake. Respondents were asked two questions adapted from the American Heart Association’s (AHA) Life’s Simple 7 score [32] and similar to brief assessments of FV intake used in other studies [17,29,33]: “How much fruit (in cups) do you eat in an average day?” and were prompted not to include fruit juice, and also were asked “How many vegetables (in cups) do you eat in an average day?” and were prompted ***not*** to include French fries to align FV intake with U.S. dietary guidelines to seek nutrient dense foods [34]. Response options were in half-cup increments from 0 to 6 cups per day. Skin carotenoids were measured three times by pressure-mediated reflection spectroscopy (RS score) (the “Veggie Meter^®^”, Longevity Link Corporation, Salt Lake City, UT, USA), a validated, non-invasive optical method to detect carotenoids in human skin [35,36,37,38]. The mean of the three measurements was calculated. Scores range from 0 to 800, with higher scores indicating a higher level of carotenoids in the skin and higher intake of FV containing carotenoids.

Finally, the survey asked participants in the FMNP education event about its impact on their knowledge, self-efficacy, and behavioral intentions. Three statements reflected distinct knowledge content embedded in the education curriculum: “I learned about a benefit program (such as SNAP/EBT or FMNP) that can be used at this farmers’ market”, “I learned a new way to cook or prepare vegetables or fruit”, and “I learned how to store my produce to prevent spoilage”. Self-efficacy was assessed with one statement about shopping for FV: “I feel more confident in selecting and purchasing produce”. Single-item assessment of self-efficacy has been found to correlate strongly with multi-item scales in the context of healthy eating and changing nutrition habits [39]. Behavioral intentions included “I plan to purchase fruits and vegetables that I do not usually buy at today’s market” and “I plan to make today’s featured recipe at home”. Participants reacted to each statement using a 4-point Likert scale from “Strongly Agree” to “Strongly Disagree” and we report the combined percentage who strongly or somewhat agreed.

### 2.2. Analysis

All continuous variables were checked for normality (skewness and kurtosis < +/−2), and six outliers (>mean + 3 SD; >USD 90/week) were removed from usual FV spending at the FM. All data were summarized with percentages and means. Characteristics and behaviors of respondents who participated in the FMNP education event were compared to non-participant shoppers with 95% confidence. Differences in percentages were identified by Chi-square tests except for one variable which had small cells sizes that necessitated the use of a Fisher’s exact test, and differences in means were identified by *t* tests assuming homogeneity of variance except for one variable when Levene’s test for equality of variance indicated that separate variance calculations were required. All analyses were performed in SPSS v25 (IBM Corp., Armonk, NY, USA).

## 3. Results

Market characteristics and information about education events were collected on 20 of the 26 days when FM shopper surveys were collected. The size of the FM on these days ranged from 1 to 8 food vendors, with a mean of 3.7 total food vendors and 2.4 FV vendors (Table 1). Five of the survey days were at a FM with one youth agriculture program as the only vendor. Of the nine FMNP lessons, three lessons were presented more than once: “Produce spotlight” (61.1%), “Eat More Fruits and Vegetables” (38.9%), and “Portion size” (16.7%). Implementation of the lessons often included food sampling alone (63.2%) or food sampling with a cooking demonstration (15.8%). Among events that featured a recipe or tasting, fruits featured were peaches (26.7%), apples (20.0%), currants and watermelon (6.7% each). Vegetables most often featured were onions and kale (60.0% each), Swiss chard and tomatoes (53.3% each), and corn, cucumbers, and summer squash/zucchini (46.7% each). In addition to the FMNP education events, most FM hosted other activities on survey days, such as food sampling (45.0%), nutrition education (25.0%), market tours (15.0%), and cooking demonstrations (5.0%).

FM shoppers were mostly female (81.5%), more than half were 50 years or older, more than half had graduated from college, and they were diverse racially (30.5% Black, 23.1% White), ethnically (36.8% Hispanic), and economically (Table 2). At the time of the survey, 25.1% received SNAP benefits and 10.3% were enrolled in WIC. There were no significant differences in the characteristics of participants and non-participant FM shoppers (data not shown).

### 3.1. Shopping and Eating Behaviors

Overall, FM shoppers reported that they usually purchased FV at the FM 2.41 times per month (SD = 1.53) and spent USD 22.99 for FV each time they shopped at a FM (SD = 14.92; Table 3). Most shoppers had used FMNP coupons (30.2%) or SNAP benefits (70.2%) at a FM at least once. On the survey day, 59% of the FM shoppers reported that they planned to purchase specific fruits or vegetables before they came to the FM; most commonly planned fruits were peaches (31.5%), apples (19.5%), and blueberries (8.0%), and vegetables were tomatoes (28.5%), corn (20.0%), carrots (18.0%), kale (17.0%), beets (16.5%), onions (11.5%), lettuce (11.0%), and summer squash/zucchini (11.0%). In total, 65.2% of FM shoppers did purchase FV on the survey day. The mean usual fruit intake among FM shoppers was 2.24 cups/day (SD = 1.38) and intake of vegetables was 2.44 cups/day (SD = 1.40), with mean skin carotenoid RS score of 314.36.

Almost all survey respondents reported that they participated in the FMNP education event (83.6%), whereas 30 reported that they did not participate (8.0%) and 32 did not respond to that question (8.5%). FM shopping frequency and money spent did not differ for participants and non-participant shoppers. Among SNAP recipients, FMNP education event participants were more likely to have ever purchased food at a FM with SNAP benefits (73.0 vs. 20.0%, *p* = 0.028), whereas among respondents eligible for FMNP coupons, education event participants and non-participants were equally likely to have purchased FV from a FM with FMNP coupons.

Education event participants and non-participant shoppers were equally likely to plan to buy specific FV before they came to the market, but participants were more likely to have purchased FV on the survey day (66.8 vs. 43.5%, *p* = 0.024). Overall, 43.8% of survey respondents purchased a ‘featured’ fruit or vegetable on the survey day, and the percentage did not differ according to education event participation. Education event participants were more likely to be finished shopping than non-participant shoppers (65.8 vs. 27.6%, *p* < 0.001).

### 3.2. Perceptions of Impact among Participants

Overall, participants perceived that the FMNP education event had a positive impact (Figure 1). Participants reported that they learned about benefits programs such as SNAP/EBT or FMNP that can be used at FM (74.2%), how to store produce to prevent spoilage (71.3%), and new methods for cooking or preparing FV (89.7%). Participants also reported that they felt more confident in selecting and purchasing produce (89.2%). Furthermore, participants reported intentions to buy FV that they do not usually buy (84.3%) and to make the featured recipe at home (92.8%).

## 4. Discussion

Overall, FM shoppers in NYC self-reported high FV consumption (about 4.5 cups FV daily), which met recommended quantities for most adults according to the Dietary Guidelines for Americans [34]. Self-reported FV consumption included dried beans and starchy tubers such as potatoes (excluding French fries) and, therefore, may not align well with World Health Organization (WHO) FV recommendations which exclude these items [40]. High self-reported FV intake was supported by high objectively measured skin carotenoid RS score (314), which also was within the healthy range (between 280 and 480) [41]. Legumes and starchy tubers contain few carotenoids [42] and, therefore, skin carotenoid measures provide strong evidence that FV intake is high with respect to WHO FV recommendations [40]. FV intake among FM shoppers in NYC was substantially higher than amounts reported among U.S. residents overall (1.0 cups of fruit and 1.7 cups of vegetables daily) [43], and higher than objective carotenoid RS scores among shoppers in supermarkets and corner stores (mean RS score = 250.5, SD = 75.4) [44]. However, this relatively high consumption of FV among FM shoppers in NYC is similar to levels reported among FM shoppers in rural North Carolina (4.3 servings) and rural Kentucky (3.7 servings) [17]. This suggests that FM shoppers irrespective of location may consume higher quantities of FV than is typical in the U.S.

We also found that FM shoppers typically shopped for FV at a FM two or three times a month and spent approximately USD 23 in money or benefits at each time. Prior research suggests that FM shopping and frequency of FM shopping is positively associated with self-reported and objectively assessed FV intake [13,14,15,16,17,18,19]. Regular FM shopping may be a habit that supports FV intake, which is an important aspect of healthy eating. However, these cross-sectional survey data cannot suggest whether FM shopping causes higher FV intake, adults with higher FV intake more often shop at the FM, or there is another explanation for observed associations between FM shopping and FV intake.

Implementation of the NYS FMNP Education Event Guide at seven FM in NYC focused on only three of the nine lessons, and incorporated food sampling two-thirds of the time. This suggests that cooperative extension educators could use these early implementation experiences to hone these lessons for dissemination, and that further research is needed to understand why the other six lessons were not selected and how they might be improved. Participation in the education events was widespread among FM shoppers from all demographic groups; and, contrary to our hypothesis, there was no higher participation in the education events among WIC recipients, older respondents, low-income households, nor FMNP coupon users. Given the relatively high levels of FV consumption among FM shoppers overall, education events would not be expected to substantially increase FV intake. Shoppers who already eat recommended quantities of FV are unlikely to make large increases in intake nor to benefit greatly from those changes if they do make them. However, approximately one-third of FMNP education events in NYC featured kale and/or Swiss chard (dark green leafy varieties that are infrequently consumed in the U.S. diet [5]), yet just 17% of FM shoppers in our sample already planned to buy kale at the FM. More frequent promotion of FV supportive of the dietary variety promoted by the Dietary Guidelines for Americans such as dark green, red, and orange vegetables [34] may support increased overall familiarity with these items as an important precursor to shopping and eating behavior change. Future research is needed to explore how education events may increase FV familiarity.

We found that education event participants were more likely to purchase any FV on the survey day than non-participant shoppers, but participants were also more likely to have finished their shopping. Non-participant shoppers may have purchased FV in the remaining time at the FM. No difference was observed in the purchase of a “featured” FV between education event participants and non-participants. This contradicts other studies that reported FM shoppers who sampled a food at the FM were more likely to want to prepare that food at home [45] and to purchase the ingredients for the recipe [46] compared to shoppers who did not sample the food. However, in the context of widespread participation in the education events (84%) and some missing data on participation (9%), we may have lacked power to detect differences in the purchase of the featured FV.

All education event participants reported positive perceptions of the impact of the education on knowledge, self-efficacy, and behavioral intentions. Together, these results suggest that FMNP education events are satisfying to participants, and may support knowledge acquisition, growth in self-efficacy and behavior change such as increased variety of FV purchased. To the extent that FMNP coupons can be used to entice new shoppers to the FM who may have inadequate FV intake, education events at FMs have the potential to support greater knowledge, confidence, and behavior change.

### Limitations

This study had several limitations that deserve note. First, respondents to the FM shopper survey were mostly women and educated (more than half held college degrees), which is consistent with one study in rural North Carolina where 43.9% of FM shoppers held a college degree [17]. However, results may not be generalizable to men, adults with less education, and those lacking familiarity with FM. Second, this study used a cross-sectional research design from which it is not possible to infer causality between participation in the education event and any shopping or eating behaviors.

Third, the shopper survey relied on brief assessment tools given the public intercept approach to data collection. The two-item assessment of FV consumption is considered valid for gross estimates of FV intake, has higher validity among women (most of this sample), is considered appropriate for resource- and time-constrained circumstances, and may over-estimate fruit intake [33]. Contrasts between self-reported FV consumption and objectively measured skin carotenoids suggest that two-item self-report of FV may be inflated in this sample. Each 100 units in the RS score corresponds to approximately one serving of FV consumed per day [47], suggesting that FV intake was somewhat more than 3 cups per day in this sample of FM shoppers. Interpretations of FV consumption should consider this measurement bias.

Fourth, assessment of knowledge, self-efficacy, and behavioral intentions also used a single or several items. Although single-item assessment of self-efficacy has been found to correlate strongly with multi-item scales in the context of healthy eating and changing nutrition habits [39], the items used in this study queried specific aspects of the FM education lessons and were not previously validated. In addition, perceptions of the impact of the education events may be overstated due to social desirability bias. Interpretations of the impact of FMNP education events on knowledge, self-efficacy, and behavioral intentions need to consider these limitations.

## 5. Conclusions

Education events at FM are a vehicle through which to provide FM shoppers with information, preparation methods, and recipes for seasonal produce; are broadly accepted by FM shoppers of all characteristics; and are perceived by FM shoppers as beneficial to their knowledge, self-efficacy, and behavioral intention. Cooperative extension educators could further focus the three piloted lessons more specifically on the dark green, red, and orange vegetables infrequently consumed in the U.S. to promote their overall familiarity, and also explore why the other six lessons were not implemented. FM shoppers generally have high self-reported and objectively measured FV consumption, suggesting FM education events may not further improve this positive dietary behavior. Future research should explore how FM education events may influence other dietary outcomes such as variety of FV consumed in longitudinal data that can examine potential causality.

## Figures and Tables

**Figure 1 nutrients-14-00436-f001:**
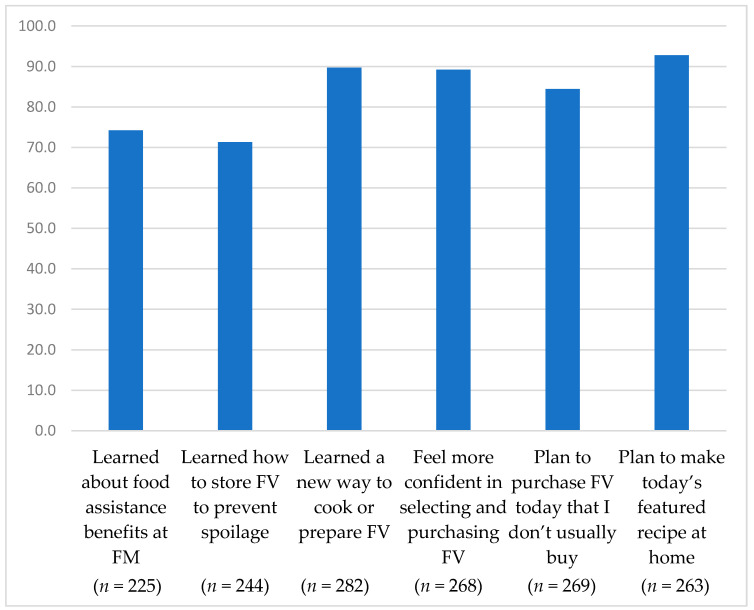
Perceptions of the impact of FMNP education events on knowledge, self-efficacy and behavioral intention.

**Table 1 nutrients-14-00436-t001:** Characteristics of farmers’ markets and education events on survey days.

**Farmers’ Markets**		**Mean**	**SD**
**Size** (***n*** **= 18**)	Number of Food Vendors	3.7	2.56
	Number of FV Vendors	2.4	1.42
**FMNP Education Events**		**Count**	**%**
**Lessons Featured** **(*n* = 18)**	1. Eat More Fruits and Vegetables	7	38.9
	2. Eating Healthy on a Budget	1	5.6
	3. Eating Healthy with Kids	0	0.0
	4. Food Safety	0	0.0
	5. Meal Planning	0	0.0
	6. Portion Size	3	16.7
	7. Sugar Sweetened Beverages	0	0.0
	8. Produce Spotlight	11	61.1
	9. Senior Nutrition	0	0.0
**Activity** **(*n* = 19)**	Food Sampling Only	12	63.2
	Cooking Demonstration & Food Sampling	3	15.8
	Neither	4	21.1
**Featured Fruits** **(*n* = 15)**	Peaches	4	26.7
	Apples	3	20.0
	Currants	1	6.7
	Watermelon	1	6.7
**Featured Vegetables** **(*n* = 15)**	Kale	9	60.0
	Onions	9	60.0
	Swiss Chard	8	53.3
	Tomatoes	8	53.3
	Corn	7	46.7
	Cucumbers	7	46.7
	Squash, summer/zucchini	7	46.7
	Carrots	5	33.3
	Peppers, sweet	5	33.3
	Garlic	3	20.0
	Beets	2	13.3
	Bok choy	2	13.3
	Collard greens	2	13.3
	Fennel	1	6.7
	Garlic scapes	1	6.7
	Green beans	1	6.7
	Green onions	1	6.7
	Peppers, hot	1	6.7
**Other Activities** **(*n* = 20)**	Food Sampling	9	45.0
	Nutrition Education only	5	25.5
	Market Tours	3	15.0
	Cooking Demonstration	1	5.0

Abbreviations: FV, fruit and vegetables; FMNP, Farmers’ Market Nutrition Program.

**Table 2 nutrients-14-00436-t002:** Survey respondent characteristics.

Characteristic		Total Survey Respondents (*n* = 377)		
		*n*	Count	%
Women		372	303	81.5
**Age**	<20 years	372	8	2.2
	20–29 years		42	11.3
	30–39 years		53	14.2
	40–49 years		66	17.7
	50–59 years		91	24.5
	60+ years		112	30.1
**Race**	American Indian	377	9	2.4
	Asian		28	7.4
	Black		115	30.5
	Native Hawaiian/Pacific Islander		2	0.5
	White		87	23.1
	Multiracial		46	12.2
	Prefer not to answer		90	23.9
Hispanic		342	126	36.8
**Highest year** **of school** **completed**	<High school	369	23	6.1
	High school graduate or GED		56	14.9
	Some college		95	25.2
	College graduate		195	51.7
**Marital status**	Single	369	168	45.5
	Living with partner		24	6.5
	Married		110	29.8
	Separated		14	3.8
	Divorced		27	7.3
	Widowed		26	7.0
**Annual household income**	<USD 20,999	359	59	16.4
	USD 21,000–39,999		66	18.4
	USD 40,000–59,999		68	18.9
	USD 60,000–79,999		34	9.5
	USD 80,000–99,999		28	7.8
	>USD 100,000		50	13.9
	Prefer not to answer		54	15.0
Received SNAP benefits at time of survey		334	84	25.1
Enrolled in WIC at time of survey		349	36	10.3

Abbreviations: GED, General Education Diploma; SNAP, Supplemental Nutrition Assistance Program; WIC, Special Supplemental Nutrition Program for Women, Infants, and Children.

**Table 3 nutrients-14-00436-t003:** Shopping and eating among FMNP education event participants and non-participant shoppers.

	Total Survey Respondents (*n* = 377)			Education Event Participants (*n* = 315)		Non-Participant Shoppers (*n* = 30)		*p*
**Usual FM Shopping**	* **n** *	**Mean**	**SD**	**Mean**	**SD**	**Mean**	**SD**	
Frequency of FV purchases at FM (times/month)	376	2.41	1.53	2.44	1.53	2.02	1.63	0.152
Usual FV spending at FM (USD/time)	329	22.99	14.92	22.68	14.65	26.72	19.28	0.316
		**Count**	%	**Count**	%	**Count**	%	
Ever used FMNP coupons ^a^	139	42	30.2	37	31.4	2	22.2	0.567
Ever used SNAP benefits at FM ^b^	84	59	70.2	54	73.0	1	20.0	0.028
**Survey Day FM Shopping**								
Planned to purchase specific FV	363	214	59.0	177	57.3	20	74.1	0.089
Purchased any FV at FM	336	219	65.2	191	66.8	10	43.5	0.024
Purchased featured FV ^c^	185	81	43.8	69	46.3	5	35.7	0.447
Finished shopping at FM	350	218	62.3	192	65.8	8	27.6	<0.001
**FV Intake**		**Mean**	**SD**	**Mean**	**SD**	**Mean**	**SD**	
Fruit intake (cups/day)	369	2.24	1.38	2.24	1.41	2.19	1.28	0.850
Vegetable intake (cups/day)	370	2.44	1.40	2.43	1.39	2.18	1.40	0.347
Mean skin carotenoids (RS score)	360	314.36	140.38	313.50	138.95	305.24	167.65	0.768

Abbreviations: FM, farmers’ market; SNAP, Supplemental Nutrition Assistance Program; WIC, Special Supplemental Nutrition Program for Women, Infants, and Children; FMNP, Farmers’ Market Nutrition Program; FV, fruit and vegetables *T*-test for differences in means; Pearson’s Chi-square and Fisher’s exact tests for differences in percentages ^a^ among shoppers either over 60 or enrolled in WIC ^b^ among shoppers enrolled in SNAP ^c^ at a FM with a “produce spotlight”, cooking demonstration, or food sampling.

## Data Availability

Data are available from the authors upon request.

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
