# Peer review of "Farmers’ Market Nutrition Program Educational Events Are Broadly Accepted and May Increase Knowledge, Self-Efficacy and Behavioral Intentions"

_nutrients, 2022, doi:10.3390/nu14030436_

Round 1
Reviewer 1 Report
Thank you for the opportunity to review this manuscript. This is an interesting topic that can be considered by readers. However, after reviewing the manuscript, there are some points I think the authors should consider in order to improve this work. These are among others:
* How was your sample size exactly calculated in your study? Have you used e.g. GPower?
* My greatest concern is that only responses to a single questions were used to measure such variables as for example "self-efficacy", "behavioural intentions" etc. How can we be sure that the results are reliable and accurate, since no psychometric indicators for these measures are available? Did the authors run any pilots to validate their measures? It reliability and validity needs to be tested.
* Why was the Student parametric t-test used? Have the relevant assumptions been checked (e.g. a normal distribution, homogeneity of variance)?
* I would recommend expanding the Discussion section as to how some of the results of the study could be implemented rather than a repeat of what was already discussed.
* In my opinion, the "Limitations" section needs to be corrected.
Author Response
Reviewer 1:
This is an interesting topic that can be considered by readers. However, after reviewing the manuscript, there are some points I think the authors should consider in order to improve this work.
- How was your sample size exactly calculated in your study? Have you used e.g. GPower?
We added detail to the methods section which describes how we arrived at our target sample size (lines 109-111).
- My greatest concern is that only responses to a single questions were used to measure such variables as for example "self-efficacy", "behavioural intentions" etc. How can we be sure that the results are reliable and accurate, since no psychometric indicators for these measures are available? Did the authors run any pilots to validate their measures? It reliability and validity needs to be tested.
We thank the reviewer for this important comment. Public intercept surveys need to be brief to successfully recruit respondents. We added details to the methods section that support the use of single-item measures in field research (lines 169-170, 187-189). However, we agree that this is a limitation of the study and discuss this further in the limitations section.
- Why was the Student parametric t-test used? Have the relevant assumptions been checked (e.g. a normal distribution, homogeneity of variance)?
We added detail to the analysis section which now describes how we checked for normality, how outliers for one variable were excluded from analysis, how homogeneity of variance was tested, and adds details to the analytic tests used (lines 195-204).
- I would recommend expanding the Discussion section as to how some of the results of the study could be implemented rather than a repeat of what was already discussed.
We thank the reviewer for this important comment. We revised and expanded the discussion section to elaborate on how the findings could be used for program implementation and suggest areas for future research (lines 299-302, 312-316, 364-367).
- In my opinion, the "Limitations" section needs to be corrected.
We thank the reviewer for this important comment. We revised and expanded the limitations section to provide more detail on the limitations of the study, how they were minimized, and how they may effect the observed results (lines 336-359).
Reviewer 2 Report
The manuscript entitled ‘Farmers’ Market Nutrition Program Educational Events are Broadly Accepted and May Increase Knowledge, Self-Efficacy and Behavioral Intentions’ presents interesting issue, however some corrections are needed
- Line 38 – ‘Potatoes’ – In many countries potatoes do count towards your 5 A Day https://www.nhs.uk/live-well/eat-well/5-a-day-what-counts/. Please comment on
- Please see WHO document https://www.who.int/dietphysicalactivity/publications/f&v_intake_measurement.pdf - the inclusion/ exclusion of potatoes, legumes, tubers should be justified and discussed
- Is table 1 necessary?
- Line 163 – ‘FV intake was assessed by self-report’ – please specify it . More information is needed about the validity and reliability of each measure. Additionally, any limitations in reliability and validity need to be addressed in the discussion.
- ‘Respondents were asked two questions adapted from the American Heart Association’s (AHA) Life’s’ – why only two questions? Authors should present here why and discuss the potential limitations of the methodology.
- Was the normality of distribution tested? The information about it should be added and authors should be consequent. If data have normal distribution, they should be treated as such, if not, nonparametric tests should be applied. Please specify it.
- Line 207 - ‘FM shoppers were mostly female (81.5%),’ – from one hand this is expected (as the person who mainly is shopping are women), but on the other hand such small share of men is bias. Please comment on and indicate as a potential limitation.
- The discussion section could be expanded. Authors should in their discussion include 3 areas: (1) compare gathered data with the results by other authors, (2) formulate implications of the results of their study and studies by other authors, (3) formulate the future areas which should be studied. Authors should present here and discuss the limitations of their study.
Author Response
Response to reviewer 2:
The manuscript entitled ‘Farmers’ Market Nutrition Program Educational Events are Broadly Accepted and May Increase Knowledge, Self-Efficacy and Behavioral Intentions’ presents interesting issue, however some corrections are needed.
- Line 38 – ‘Potatoes’ – In many countries potatoes do count towards your 5 A Day https://www.nhs.uk/live-well/eat-well/5-a-day-what-counts/. Please comment on. Please see WHO document https://www.who.int/dietphysicalactivity/publications/f&v_intake_measurement.pdf – the inclusion/ exclusion of potatoes, legumes, tubers should be justified and discussed
We thank the reviewer for this important comment. We included information on differences between U.S. and WHO dietary recommendations, and how our objective measurement of carotenoids provides data supportive of either recommendation (lines 274-281).
2. Is table 1 necessary?
We agree that table 1 is not necessary and have omitted it.
3. Line 163 – ‘FV intake was assessed by self-report’ – please specify it . More information is needed about the validity and reliability of each measure. Additionally, any limitations in reliability and validity need to be addressed in the discussion.
We thank the reviewer for this important comment. We added details to the description of tool used for self-report of FV intake in the methods section, and also fully discuss the potential limitations of this tool in the limitation section (lines 169-170, 342-351). Further details were also added to the discussion section about how the self-report and objective measures of FV intake support the finding of adequate FV intake among FM shoppers overall (lines 293-296).
4. ‘Respondents were asked two questions adapted from the American Heart Association’s (AHA) Life’s’ – why only two questions? Authors should present here why and discuss the potential limitations of the methodology.
Please see our response to the prior comment.
5. Was the normality of distribution tested? The information about it should be added and authors should be consequent. If data have normal distribution, they should be treated as such, if not, nonparametric tests should be applied. Please specify it.
We added detail to the analysis section which describes how we checked for normality, how outliers for one variable were excluded from analysis, how homogeneity of variance was tested, and adds details about the analytic tests used (lines 195-204).
6. Line 207 – ‘FM shoppers were mostly female (81.5%),’ – from one hand this is expected (as the person who mainly is shopping are women), but on the other hand such small share of men is bias. Please comment on and indicate as a potential limitation.
We added detail to the limitations section that discusses how results may not be generalizable to men (lines 336-338).
7. The discussion section could be expanded. Authors should in their discussion include 3 areas: (1) compare gathered data with the results by other authors, (2) formulate implications of the results of their study and studies by other authors, (3) formulate the future areas which should be studied. Authors should present here and discuss the limitations of their study.
We thank the reviewer for this important comment. We revised and expanded the discussion section to make additional comparisons between this study and other results, elaborate on how the findings could be used for program implementation, and suggest areas for future research (lines 290-292, 299-302, 321-324, 364-371). We also revised and expanded the limitations section to provide more detail on the limitations of the study, how they were minimized, and how they may affect the observed results (lines 336-359).
Round 2
Reviewer 2 Report
I appreciate the great efforts that the authors have made in response to my questions and concerns. I have no further comments. Congratulation for the authors this is a very nice article.